# URB3D
## Metoda trójwymiarowego modelowania obszarów urbanistycznych z wykorzystaniem metod fotogrametrii



**Autorzy**: Daniel Borkowski◉ · Julia Farganus◉ · Rafał Mielniczuk◉ · Katarzyna Wochal◉

**Opiekun:** Marek Krótkiewicz

### Streszczenie

Przedmiotem projektu było wykonanie aplikacji wykorzystującej metody fotogrametrii do modelowania trójwymiarowych scen miejskich. Zaimplementowany program umożliwia użytkownikowi zrekonstruowanie chmury punktów i modelu 3D na podstawie podanych na wejściu zdjęć fotogrametrycznych za pomocą dostosowanych do specyfiki problemu w toku eksperymentów rozwiązań *structure from motion* i *gaussian splatting*. Otrzymana w ten sposób chmura może być przez użytkownika poddana segmentacji semantycznej, przeprowadzanej przez wytrenowany do tego celu model sztucznej inteligencji w postaci sieci neuronowej. Aplikacja oferuje również wizualizację wykonanych obliczeń, która możliwa jest dzięki użyciu przystosowanego dla większej wydajności mechanizmu renderowania.

Wytworzony produkt informatyczny ze względu na integrację wielu rozwiązań i efektywną implementację procesu *end-to-end* przejawia potencjał w zastosowaniach biznesowych począwszy od branż takich jak gry wideo, przez architekturę, robotykę, pojazdy autonomiczne, skończywszy na modelowaniu urbanistycznym.

## 1  WPROWADZENIE

Rekonstrukcja, klasyfikacja i wizualizacja scen urbanistycznych to dynamicznie rozwijające się zagadnienie, które zyskało na znaczeniu dzięki rosnącej dostępności nowoczesnych technologii, takich jak LiDAR, oraz postępowi w dziedzinie sztucznej inteligencji. Kluczowym wyzwaniem pozostaje jednak efektywne przetwarzanie ogromnych zbiorów danych – chmur punktów, które nierzadko obejmują miliony elementów. Choć na rynku istnieją liczne programy i algorytmy wspierające tego typu analizy, ich skuteczne wykorzystanie w praktyce bywa problematyczne, głównie z uwagi na skalę i złożoność danych urbanistycznych.

Rozwiązanie będzie umożliwiało przeprowadzenie rekonstrukcji do modelu trójwymiarowego na podstawie odpowiednio przygotowanego zbioru zdjęć, klasyfikację otrzymanej sceny na zbiór pre-definiowanych klas istotnych w kontekście urbanistycznym, oraz wizualizację wykonanych obliczeń.

Poniżej znajdują się cele, które zostaną zrealizowane w przedsięwzięciu:

1. skomponowanie własnego zbioru danych,

2. wykorzystanie algorytmu Gaussian Splatting do rekonstrukcji sceny 3D,

3. filtracja chmury punktów przy użyciu różnych technik,

4. zastosowanie architektur sieci neuronowych takich jak PointNet do klasyfikacji chmury punktów,

5. adaptacja istniejących bibliotek do wizualizacji wyników,

6. implementacja własnego algorytmu do renderowania gaussianów,

## 2  STAN WIEDZY

Unikalność projektu wynika z połączenia wielu rozwiązań które istnieją samodzielnie na rynku. Algorytm Structure-from-Motion [10] jest popularną fotogrametryczną techniką uzyskiwania chmury punktów ze zbioru zdjęć i jego implementacja oferowana jest m. in. przez oprogramowanie COLMAP.

W przypadku modelu 3D często stosowaną techniką są siatki, ale ich wadą jest niekompatybilność z algorytmami sztucznej inteligencji. Popularne są też rozwiązania wykorzystujące sieci neuronowe jak np. NeRF [7], jednak długi czas trenowania, osiągający nawet parę dni, jest nieefektywny. Z tego powodu zdecydowano się na wykorzystanie algorytmu Gaussian Splatting [4], który buduje model sceny z tzw.

gaussianów, które można interpretować jako punkty rozmyte. Istniejące adaptacje do skali urbanistycznej tego algorytmu to np. CityGaussian [6].

Ważnym krokiem jest również filtracja chmury punktów [2] w celu usunięcia odstających punktów lub tych nieistotnych dla wyników klasyfikacji. W tym obszarze znajdują się np. techniki statystyczne czy oparte na sąsiedztwie.

Istniejące architektury sieci neuronowych dla zadania segmentacji są głównie przeznaczone dla scen zamkniętych lub pojedynczych obiektów. Popularnym rozwiązaniem jest PointNet [8] oraz jego następnik PointNet++ [9] oparte na wielowarstwowym perceptronie, jak i również bardziej skomplikowane rozwiązania jak KPConv [12] wykorzystujące konwolucje. Wyzwaniem dla projektu jest dostosowanie takich architektur do chmury punktów o wielkości rzędu milionów punktów.

W przypadku renderowania istnieją rozwiązania przeznaczone zarówno do chmur punktów jak i do splatów, zaimplementowane często przy pomocy WebGL, jak [5]. Wyzwanie stanowi jednak wydajne i efektywne przedstawienie milionów elementów, co wymaga skorzystania z GPU i niskopoziomowego pisania kodu.

## 3 WYNIKI

### 3.1 Wytworzone oprogramowanie

Modularność projektu sprawia, że wygodniej jest omówić niezależnie każdą z części. Dla jasności, przebieg całego procesu jest przedstawiony na schemacie 1. Każda funkcjonalność, jak i wizualizacje wyników poszczególnych zadań są dostępne poprzez interfejs użytkownika.

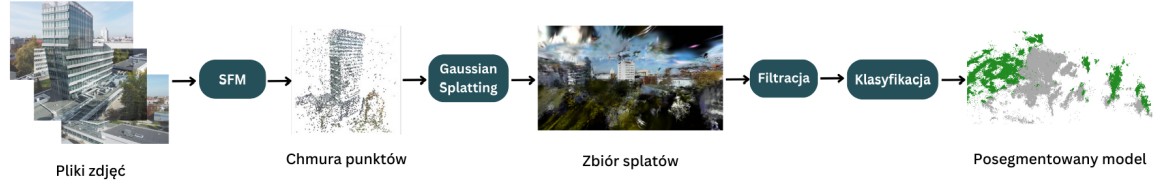

Rysunek 1: Przebieg procesu

### 3.2 Akwizycja danych

W projekcie założono wykorzystanie metod fotogrametrycznych do tworzenia trójwymiarowych modeli obszarów urbanistycznych. Za część projektu przyjęto z tego względu również pozyskanie własnych zestawów danych fotograficznych (fotogramów), które spełniałyby wymogi techniczne, umożliwiające późniejszą rekonstrukcję 3D. Niezbędne było wykonanie dużej liczby ujęć, obejmujących wiele kątów i perspektyw oraz zapewnienie odpowiedniego nakładania się zdjęć dla poprawnego działania oprogramowania fotogrametrycznego, które identyfikuje i dopasowuje wspólne punkty widoczne na wielu zdjęciach.

Akwizycję zrealizowano na kampusie **Politechniki Wrocławskiej**, koncentrując się na budynkach **C5**, **C7** oraz Strefie Kultury Studenckiej (**SKS**) 2 i pozyskując zdjęcia zarówno z lotów bezzałogowym statkiem powietrznym, jak i z poziomu gruntu. Stanowią one kompletne zbiory danych, które spełniły wymogi jakościowe i posłużyły do budowy testowych modeli.

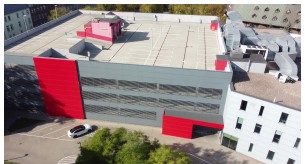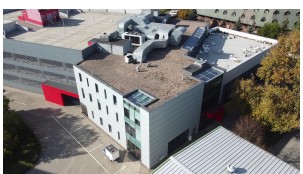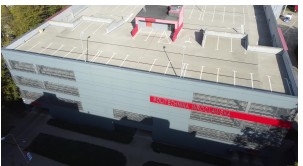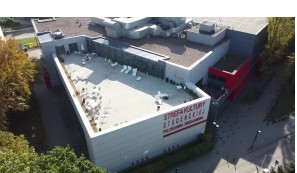

Rysunek 2: Przykładowe zdjęcia z akwizycji danych przedstawiające SKS

### 3.3   Structure from motion

Kolejnym etapem projektu było wykorzystanie techniki *Structure from Motion* (SfM) do wyznaczania struktur przestrzennych scen na podstawie dobranych zestawów zdjęć dwuwymiarowych.

Algorytmy SfM, identyfikując i łącząc punkty wspólne między zdjęciami, ustalają zarówno rozmieszczenie tych punktów w przestrzeni, jak i pozycje i orientacje kamer, z których wykonano zdjęcia. Proces ten pozwala na oszacowanie struktury trójwymiarowej sfotografowanego obszaru, czyli wygenerowanie chmury punktów odwzorowującej scenę w postaci zbioru punktów 3D o przypisanych kolorach, tak jak to pokazuje ilustracja 3.

Do realizacji tego zadania użyto popularnego narzędzia COLMAP [11] [10], a konkretnie jego wersji w formie biblioteki *pycolmap* [1], oferującej funkcjonalności m.in. do wykrywania charakterystycznych cech na obrazach, łączenia punktów wspólnych na zdjęciach czy przeprowadzania rekonstrukcji sceny 3D na podstawie dopasowań między nimi.

Uzyskana chmura punktów jest dodatkowo poddawana filtracji z wykorzystaniem metod opartych na analizie sąsiedztwa każdego punktu. Proces ten pozwala na eliminację szumów poprzez usunięcie punktów, które nie wpisują się w lokalne wzorce przestrzenne. Tak przygotowana trójwymiarowa reprezentacja sceny służy za podstawę do modelowania z zastosowaniem algorytmu Gaussian Splatting.

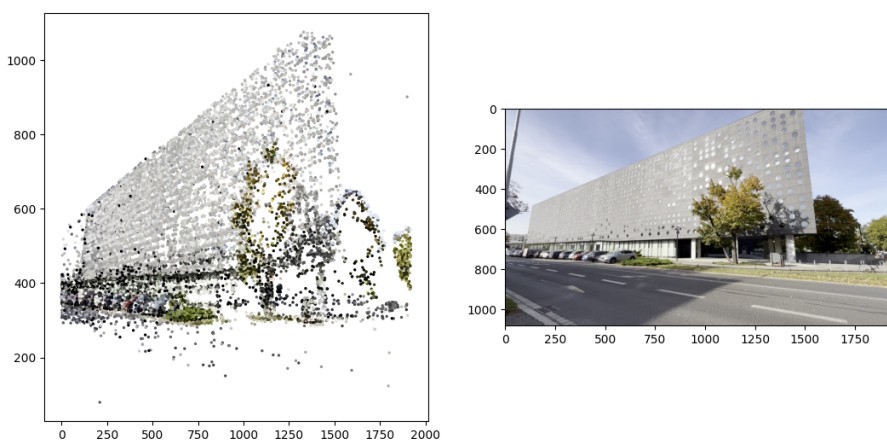

Rysunek 3: Projekcja przykładowej chmury punktów na płaszczyznę porównana do zdjęcia

### 3.4   Gaussian Splatting

Przy pomocy biblioteki *gsplat* [13] zawierającej implementację *Gaussian Splatting* w Pythonie wykonane zostały eksperymenty polegające na uruchomieniu algorytmu dla różnych wartości hiperparametrów w celu znalezienia wartości które prowadzą do jak najbardziej optymalnego procesu trenowania w kontekście czasu trwania i wykorzystania pamięci.

Na wejściu algorytmu podawana jest chmura punktów, która jest bazą do dalszego dzielenia i powstawania gaussianów, a ich parametry: pozycja, kolor, skala i rotacja są optymalizowane przy pomocy metody spadku wzdłuż gradientu. Metryki przyjęte do oceny jakości to **SSIM** (Structural Similarity Index Measure), **PSNR** (Peak Signal-to-Noise Ratio) oraz **LPIPS** (Learned Perceptual Image Patch Similarity).

Wykonane eksperymenty pokazały, że najważniejsze parametry dla przebiegu procesu to:

1. liczba Gaussianów: w przypadku scen urbanistycznych w celu oddania odpowieniej szczegółowości potrzebne jest kilka milionów Gaussianów,

2. strategia i częstość adaptacji: określają w jaki sposób oraz jak często dodawane i usuwane są Gaussiany,

3. liczba iteracji: zwykle im dłużej trenowana jest scena tym wyniki są lepsze, jednak zależy to również od przyjętej strategii. Liczba ta wpływa bezpośrednio na czas trenowania, powinna wynieść nie mniej niż kilkanaście tysięcy,

4. stopień zmiennych harmonicznych: wyrażają one kolor, im większy stopień tym lepsza jakość sceny, ale też zwiększone zużycie pamięci i wydłużony czas trenowania.

Ilustracje 4, 5 oraz 6 przedstawiają przykładowe wizualizacje (od lewej do prawej: prawdziwe zdjęcie i widok modelu), a tabela 1 zawiera wybrane wielkości dla scen testowych. Renderowania zostały wykonane przy pomocy biblioteki *nerfview* która służy do wizualizacji splatów.

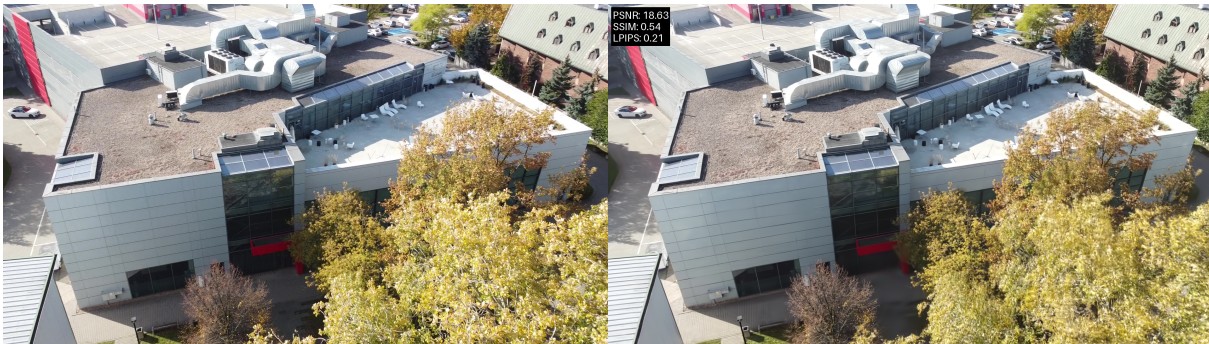

Rysunek 4: Scena SKS

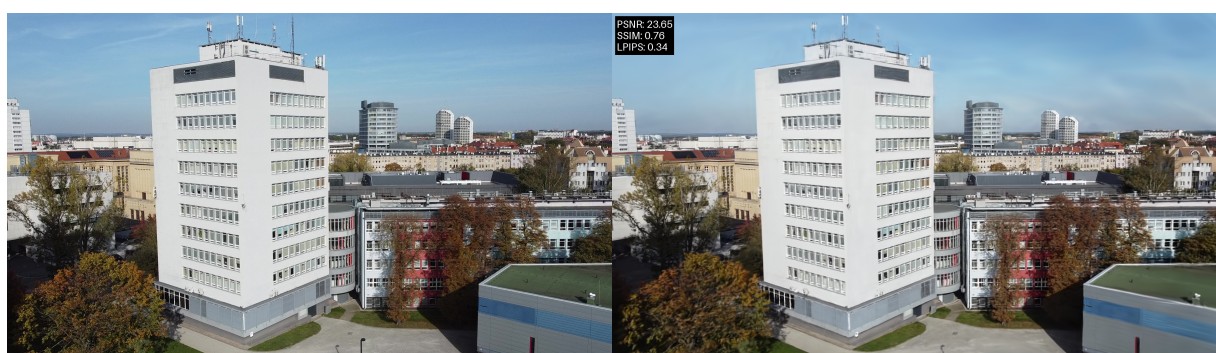

Rysunek 5: Scena C5

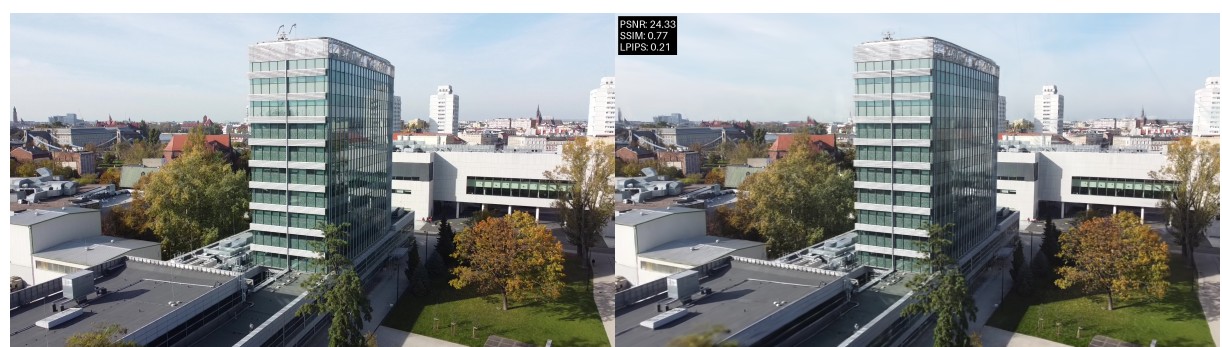

Rysunek 6: Scena C7

| scena | PSNR | SSIM | LPIPS | liczba gaussianów | czas trenowania | pamięć pliku (MB) |
|-------|------|------|-------|-------------------|-----------------|-------------------|
| SKS | 22.03 | 0.71 | 0.25 | 2,937,549 | 2h53m | 661 |
| C5 | 21.98 | 0.71 | 0.26 | 4,564,464 | 10h40m | 675 |
| C7 | 22.63 | 0.72 | 0.29 | 3,000,000 | 15h15m | 675 |

Tabela 1: Całościowe metryki dla testowych scen. Otrzymane wartości PSNR, SSIM oraz LPIPS zwykle świadczą o dobrej jakości scenie, która oddaje wystarczające szczegóły i wygładzone artefakty.

## 3.5 Segmentacja semantyczna

Otrzymana w wyniku poprzednich etapów chmura punktów poddawana procesowi segmentacji semantycznej, czyli przypisaniu każdemu z punktów odpowiedniej kategorii semantycznej opisującej obiekt, w skład którego wchodzi. Do wybranych (na podstawie popularnych w literaturze zbiorów danych służących za punkt odniesienia w testowaniu modeli) kategorii semantycznych należą między innymi *budynek*,

*droga*, czy też *zieleń miejska*. Używając biblioteki *PyTorch* do uczenia głębokiego, w oparciu o istniejące rozwiązania i aktualny stan wiedzy, przygotowano i wytrenowano własne modele *sieci neuronowych* do segmentacji semantycznej. W procesie eksperymentowania z różnymi architekturami i sposobami implementacji procesu treningu i predykcji za kluczowe pod względem wpływu na osiągi otrzymanego modelu należy uznać:

1. próbkowanie - przy przetwarzaniu zbiorów danych, w przypadku których określenie porządku jest z punktu widzenia efektywności rozwiązania bezcelowe, a które jednocześnie z punktu widzenia modelu mogą osiągać różne rozmiary, ważnym elementem procesu zarówno treningu, jak i predykcji jest odpowiednie próbkowanie całego zbioru. Jest to niezbędne ze względu na architekturę sieci neuronowych, która zakłada stały rozmiar wejścia do modelu. W obrębie tego problemu należy wyróżnić następujące czynniki:

    (a) rozmiar próbki - zbyt mały może uniemożliwić uchwycenie zależności pomiędzy zbliżonymi do siebie w chmurze punktami,

    (b) sposób próbkowania - wpływa na zależność uchwyconych w wyniku procesu uczenia wzorców od bardziej lub mniej odległych od siebie punktów. Może prowadzić do swego rodzaju zdominowania segmentowanych punktów przez jedną kategorię.

2. niezbalansowany zbiór danych - w przypadku rozpatrywania dużych scen miejskich naturalnym jest pojawienie się mniej (*samochody*, *tory kolejowe*) i bardziej (*budynki*) popularnych kategorii semantycznych. Jest to klasyczny problem uczenia maszynowego na niezbalansowanym zbiorze danych, który, niezaadresowany, prowadzi do dominacji zbioru przez punkty popularniejszych kategorii, w efekcie przekładając się na słabszą generalizację otrzymanego modelu, w szczególności dla mniej popularnych kategorii semantycznych. W toku prac rozważano dwa sposoby radzenia sobie z tym problemem:

    (a) próbkowanie - opisane wyżej,

    (b) ważenie funkcji straty - *karze* model za omijanie mniej popularnych kategorii semantycznych. Technika ta może prowadzić do nadreprezentacji tych kategorii w otrzymanym w wyniku predykcji zbiorze, jednakże przy odpowiedniej implementacji nieco słabsze wartości metryk dla popularnych kategorii są *nomen omen* balansowane przez lepsze wyniki na niedoreprezentowanych w zbiorze treningowym kategorii, prowadząc w efekcie do lepszych wartości metryk dla całego zbioru testowego.

3. dobór danych treningowych, walidacyjnych i testowych - typowy problem dla uczenia maszynowego. Dbając o generalizację modelu nie możemy dopuścić do *wycieku danych*, tj. sytuacji, w której obiecujące wyniki są spowodowane nie ową generalizacją, a pewnego rodzaju pokrewieństwem danych służących do treningu modelu i jego oceny w trakcie tego procesu lub po jego zakończeniu. Rozwiązaniem oprócz odpowiedniego podziału i doboru danych jest użycie technik przetwarzania chmur punktów związanych np. z ich obracaniem lub skalowaniem, tak aby wyuczone wzorce podlegały jak najlepszemu uogólnieniu.

Do trenowania posłużyliśmy się zbiorem *SensatUrban* [3]7, często używanym w literaturze do oceny modeli segmentacji semantycznej.

| WL | TS | OwAcc | OwF | budAcc | budkF | zieAcc | zieF | parAcc | parF |
|-----|------------|-------|-------|--------|-------|--------|------|--------|------|
| Nie | 238 950 | 43.37 | 37.50 | 83 | *87* | 0 | 0 | *51* | 11 |
| Nie | 19 509 208 | 31.68 | 30.57 | 52 | 54 | 13 | 2 | 15 | 7 |
| Tak | 238 950 | *50.69* | 45.57 | *91* | 84 | *42* | *38* | 44 | *28* |
| Tak | 19 509 208 | 31.72 | 17.89 | 52 | 24 | 13 | 19 | 15 | 15 |

Tabela 2: Metryki dla eksperymentów z modelami *PointNet*. Ważenie funkcji straty, rozmiar zbioru testowego, całkowita ważona dokładność, całkowity ważony F-score, dokładność i F-score dla kategorii: *budynek*, *teren zielony* i *parking*.

Otrzymany w wyniku tego procesu eksperymentowania model wdrożono w celu jego używania na *niewidzianych* przez niego dotychczas danych.

## 3.6  Wizualizacja

W celu zapewnienia użytkownikowi końcowemu zintegrowanego i spójnego środowiska wizualizacji całego procesu – od wgrania plików wejściowych po interakcję z modelem – zaprojektowano od podstaw interfejs oraz system renderowania.

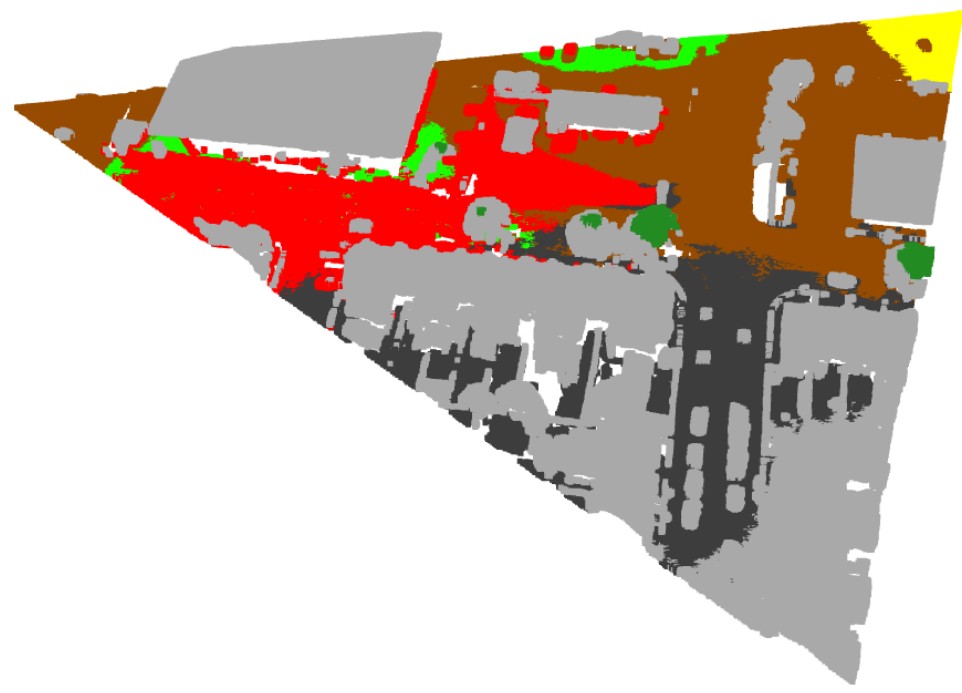

Rysunek 7: Posegmentowany zbiór testowy 238 950 punktów.

Założeniem projektu była implementacja intuicyjnego, dynamicznego i responsywnego **interfejsu**[8] przy pomocy biblioteki *PyQt* oraz języka *QML*. Interfejs został zintegrowany z wydajnym systemem **renderingu**[8] GPU, wykorzystującym technologie *OpenGL*, *OpenCL* oraz język *C*. Dodatkowo, użytkownik ma możliwość alternatywnego renderowania z wykorzystaniem biblioteki *VisPy*.

Projekt rozwiązuje problem fragmentaryczności funkcji dostępnych w innych aplikacjach, oferując spójne środowisko do obsługi modeli 3D, obejmujące procesy tworzenia, modyfikacji, segmentacji oraz wizualizacji danych.

**Rendering**

Rendering wykorzystuje plik .ply jako dane wejściowe do wczytania splatów. Splaty te są reprezentowane przez sześciany z dodatkowymi parametrami przechowywanymi w Shader Storage Buffer Object (SSBO), co umożliwia efektywny odczyt dużych ilości danych. Takie podejście jest szczególnie przydatne w przypadku scen zawierających nawet do dwóch milionów obiektów.

Do przechowywanych parametrów należą:

- pozycja,

- skala,

- rotacja,

- kolor,

- przezroczystość.

Podczas procesu renderowania, model sześcianu jest odpowiednio przekształcany na podstawie tych parametrów, co pozwala na uzyskanie splatów na wyjściu. Takie podejście umożliwia abstrakcyjne definiowanie splatów przy jednoczesnym zachowaniu wysokiej dokładności wizualnej.

## 4   PODSUMOWANIE

Rekonstrukcja i klasyfikacja krajobrazów urbanistycznych ma wiele potencjalnych zastosowań w dziedzinach takich jak *Smart City* czy też *Virtual Reality*. Projekt "urb3d"pokazał, że wykonanie takiego oprogramowania jest możliwe przy pomocy integracji istniejących rozwiązań i ich adaptacji.

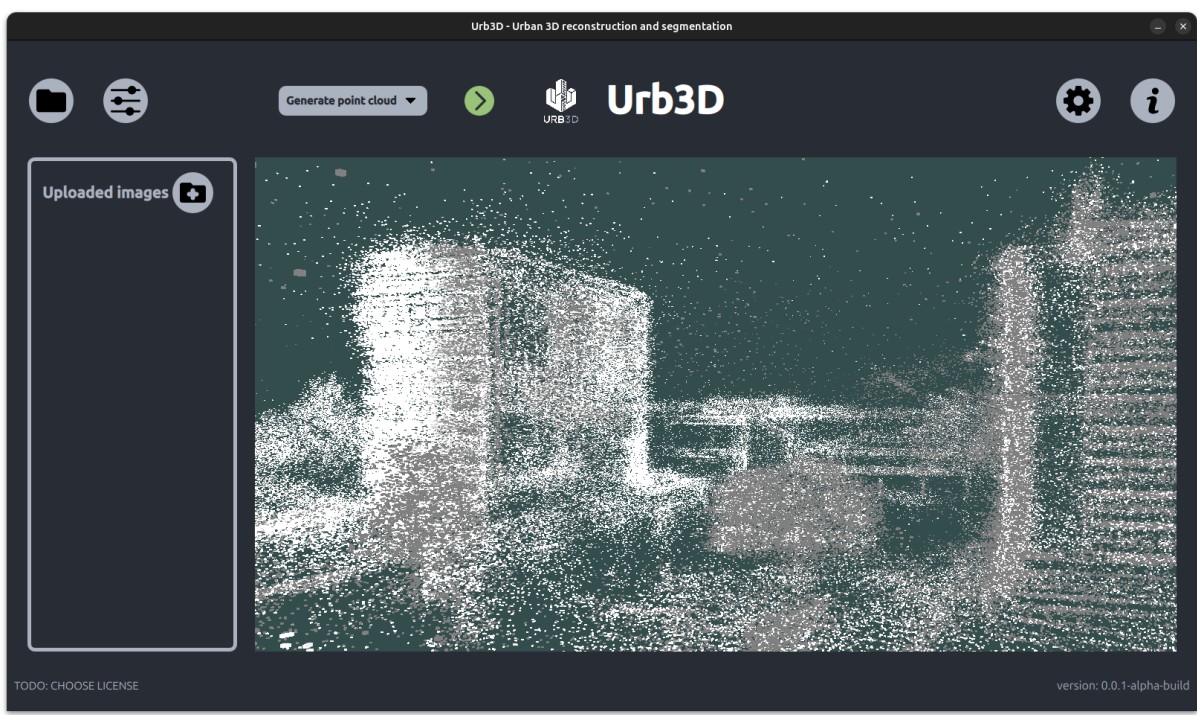

Rysunek 8: Zrzut ekranu przedstawiający główny widok aplikacji, w tym własny renderer

## 4.1 Wnioski

Zaprojektowane oprogramowanie zapewnia intuicyjne korzystanie z funkcjonalności takich jak dostosowywanie parametrów, wczytywanie zdjęć, uruchamianie poszczególnych etapów oraz przeglądanie rezultatów. Odpowiednio dobrane algorytmy zapewniają jakościowe wyniki, które mogą być dalej wykorzystane razem lub oddzielnie.

## 4.2 Kierunki rozwoju

W przyszłości projekt mógłby obejmować rekonstrukcję jeszcze większych obszarów urbanistycznych, co wiązałoby się z koniecznością zastosowania bardziej zaawansowanych technik optymalizacji. Warta wprowadzenia mogłaby okazać się kompresja danych w celu zmniejszenia końcowej liczby gaussianów w algorytmie *splattingu*. Inny kierunek rozwoju to trenowanie modelu na różnych poziomach szczegółowości (ang. Level of Detail - LoD). Otwartym rozdziałem jest też testowanie kolejnych modeli segmentacji semantycznej i ich dostrajanie.

## 4.3 Podziękowania

Szczególne podziękowania należą się operatorce drona Paulinie, bez której z pewnością sukces w obszarze akwizycji danych nie byłby możliwy.

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
