# OpenReview forum: "Metoda trójwymiarowego modelowania obszarów urbanistycznych z wykorzystaniem metod fotogrametrii"
_pwr.edu.pl/Wrocław_University_of_Science_and_Technology/2024/ZPI_Day — Wrocław University of Science and Technology 2024 ZPI Day Submission_

### Official Review · Reviewer_H5nA · 2024-12-04
**Przedmiotem projektu było opracowanie aplikacji bazującej na  metodach fotogrametrii do modelowania trójwymiarowych scen miejskich.  Uzyskany produkt charakteryzuje się sporym potencjałem aplikacyjnym  w obszarze gier video, architektury, pojazdów autonomicznych itd.**

**Confidence:** 4
**Significance Of Results:** 4
**Overall Quality:** 4

**Compliance With Template:**

4: High Quality – The article contains all the required sections, which are well-written and substantively correct, although minor errors or shortcomings may be present. The overall structure is clear and coherent.

**Description Of Results:**

4: High Quality – The results are described in detail and supported by usage examples or evaluations. The description is reliable but may lack full depth of analysis.

**Feedback On Consistency:**

Opis projektu jest spójny i zawiera wszystkie konieczne elementy takie jak: opis stanu wiedzy obszaru, którego dotyczy projekt, wykorzystane metody i narzędzia, uzyskane wyniki, podsumowanie i wnioski, kierunki dalszych prac.
Pojawiły się pewne niezręczności i niedokładności w samym opisie. Przykładowo ...do jak najbardziej optymalnego procesu trenowania...
W optymalizacji nie ma takiego pojęcia-najbardziej optymalnego.

**Potential For Development:**

Opracowana aplikacja może byś wykorzystana w wielu obszarach. Do najważniejszych z nich należą: gry wideo, modelowanie urbanistyczne czy pojazdy autonomiczne.

**Project Nature Evaluation:**

Projekt niewątpliwie zawiera elementy pracy inżynierskiej - wykorzystanie różnorodnych technik akwizycji danych, wyznaczanie struktur przestrzennych, dokonanie segmentacji semantycznej, itd. Dodatkowo opracowano interfejs użytkownika i system renderowania.

**Technical Language Precision:**

3: Average Quality – The language is mostly appropriate but may contain minor terminological or stylistic errors. Some statements might lack precision or require improvement for better readability.

---

### Official Review · Reviewer_N9sD · 2024-12-06
**Projekt wymagający zdobycia dużych kompetencji technicznych w relatywnie wąskim obszarze, nie będącym w programie studiów.**

**Confidence:** 5
**Significance Of Results:** 5
**Overall Quality:** 5

**Compliance With Template:**

5: Very High Quality – The article contains all the required sections, which are written in a very detailed, clear, and error-free manner. The structure is professional and meets expectations, and the content adheres to the highest substantive and formal standards.

**Description Of Results:**

5: Very High Quality – The results are described in detail, clearly and comprehensively, supported by thorough evaluation, analysis, and convincing usage examples. The description meets the highest substantive standards.

**Feedback On Consistency:**

Opracowanie jest napisane w sposób spójny wewnętrznie i logicznie. Rezultaty zostały bogato zilustrowane i prawidłowo skomentowane.
 W pracy przedstawiono tło technologiczne, które samo w sobie stanowi pewne wyzwanie ze względu na specyfikę projektu.

**Potential For Development:**

Rezultaty posiadają duży potencja praktycznego zastosowania w wielu obszarach zastosowań.

**Project Nature Evaluation:**

Projekt posiada charakter inżynierski i polega na wytworzeniu szeregu narzędzi w celu uzyskania efektu w postaci spójnego systemu realizującego zadania z obszaru fotogrametrii.

**Technical Language Precision:**

5: Very High Quality – The language is entirely appropriate for a technical report. All terms are used correctly and precisely, and the style is professional, clear, and coherent, without any errors or ambiguities.

---

### Official Review · Reviewer_uc79 · 2024-12-09
**The review of Metoda trójwymiarowego modelowania obszarów urbanistycznych z wykorzystaniem metod fotogrametrii**

**Confidence:** 5
**Significance Of Results:** 5
**Overall Quality:** 5

**Compliance With Template:**

5: Very High Quality – The article contains all the required sections, which are written in a very detailed, clear, and error-free manner. The structure is professional and meets expectations, and the content adheres to the highest substantive and formal standards.

**Description Of Results:**

5: Very High Quality – The results are described in detail, clearly and comprehensively, supported by thorough evaluation, analysis, and convincing usage examples. The description meets the highest substantive standards.

**Feedback On Consistency:**

The paper is straightforward and describes an interesting system focused on running image processing methods and rendering their results. The description is correct and gives the reader the opportunity to get to know their system. The language is sometimes not precise and requires some style polishing.

**Potential For Development:**

The project has potential for development however needs more evaluation in different urban scenarios. Some hyperparameters of the method have been adjusted for proof-of-concept solution and maybe require some optimization.

**Project Nature Evaluation:**

The  project is of scientific nature. The description focuses more on research aspects and their results, presenting subsequent stages of activities perforrmed in the pipeline of image and signal processing. The paper ends iwth presentation of their engineering works, presenting the technology stack. The level of utility is high in terms of investigating usage of AI models in the domain of photogrammetry and image processing.

**Technical Language Precision:**

4: High Quality – The language is appropriate for a technical report. Terminology is used correctly, and statements are precise, with only minor shortcomings that do not affect the overall clarity.

---

### Decision · Program_Chairs · 2024-12-10

Accept (Oral)